# Exposure to Ochratoxin A from Rice Consumption in Lebanon and United Arab Emirates: A Comparative Study

**DOI:** 10.3390/ijerph191711074

**Published:** 2022-09-04

**Authors:** Hussein F. Hassan, Alissar Abou Ghaida, Abeer Charara, Hani Dimassi, Hussein Faour, Rayan Nahouli, Layal Karam, Nisreen Alwan

**Affiliations:** 1Nutrition Program, Natural Sciences Department, Lebanese American University, Beirut P.O. Box 13-5053, Lebanon; 2School of Pharmacy, Lebanese American University, Byblos P.O. Box 36, Lebanon; 3Biology Program, Natural Sciences Department, Lebanese American University, Beirut P.O. Box 13-5053, Lebanon; 4Human Nutrition Department, College of Health Sciences, QU Health, Qatar University, Doha P.O. Box 2713, Qatar; 5College of Health Sciences, Abu Dhabi University, Abu Dhabi 59911, United Arab Emirates

**Keywords:** rice, ochratoxin A, ELISA, exposure, contamination

## Abstract

Our study aims to evaluate the ochratoxin A (OTA) in rice marketed in Lebanon and the United Arab Emirates (UAE), and to determine the exposure to OTA from rice consumption. All brands available in the market were collected twice (total number of collected samples: 105 and 127 in Lebanon and the UAE, respectively). Using ELISA, the OTA in 56 (53%) samples in Lebanon and 73 (58%) samples in the UAE were above the limit of quantification (0.8 μg/kg). The average concentrations of the positive samples ± standard deviations were 1.29 ± 0.32 and 1.40 ± 0.42 μg/kg in Lebanon and the UAE, respectively. Only one sample (1%) in Lebanon had a level at the borderline of the European Union (EU) limit, and two samples (1.6%) in the UAE had a level above the EU limit (5 μg/kg). The OTA in brown rice was higher than in white and parboiled rice for both countries, yet the difference was not significant. The packing season, packing country, and country of origin did not have any significant effects. The presence of a food safety certification resulted in lower OTA in the rice, but the difference was significant (*p* = 0.04) in the UAE only. Long grains had higher OTA than short grains, yet the difference was only significant in Lebanon (*p* = 0.046). The exposures were calculated as 1.27 ng/kg body weight/day in Lebanon and 1.42 ng/kg body weight/day in the UAE, and no health risk was observed for both the neoplastic and non-neoplastic effects.

## 1. Introduction

Mycotoxins are harmful secondary metabolites that are produced by different fungi, such as *Aspergillus*, *Penicillium*, *Alternaria*, *Fusarium*, and *Claviceps* [1]. While there are over 300 known mycotoxins, 6 of them are commonly detected in food, causing unpredictable and persistent food safety issues worldwide. Many crops, including nuts, cereals, fruits, and vegetables, pose a high contamination risk with mycotoxins [2]. Furthermore, if animals consume contaminated feed, mycotoxins can be detected in animal-derived foods, such as eggs, milk, and meat [3]. Contaminated food crops with mycotoxins constitute around 25% of crops worldwide [4]. Poor agricultural and harvesting practices and incorrect handling promote fungal growth, increasing the likelihood of mycotoxin formation [5,6]. Furthermore, water activity between 0.80 and 0.99 and temperatures between 25 and 30 °C promote fungal growth and thus increase the chances of mycotoxin presence [7]. Mycotoxins pose a risk to human health because they can cause severe and irreversible harm, such as cancer [8].

Among the mycotoxins are ochratoxins, which are secondary metabolites generated by *Aspergillus circumdati*, *Aspergillus niger*, *Penicillium verrucosum*, and *Penicillium nordicum.* There are three types of ochratoxins: ochratoxin A, ochratoxin B, and ochratoxin C, with ochratoxin A (OTA) being the most common and poisonous type, of which kidney injury is the most sensitive and noticeable outcome [9]. OTA can induce cell apoptosis and it inhibits protein and RNA synthesis. Different studies have found that OTA exposure causes an accumulation of free radicals [10]. Presently, OTA is classified as a possible human carcinogen (group 2B) by the International Association for Research on Cancer (IARC), despite the fact that many researchers have proposed classifying it as 2A instead [11].

Rice is one of the world’s most crucial food crops, with more than 90% production in tropical and semitropical Asia. According to the Food and Agriculture Organization (FAO), rice is grown in 113 countries, and it is the staple food for most of the world’s population. Rice provides 27% of the dietary energy and 20% of the dietary protein in developing countries [12,13]. Rice can become infested with fungus if storage conditions do not meet food safety standards, resulting in the loss of this staple grain and, as a result, a detrimental impact on the economies of rice-producing countries [14]. Each year, 15% of the produced rice is wasted primarily on parasite contamination and other potentially harmful species that commonly arise under poor storage conditions. As a result, and particularly in Asia, mycotoxin contamination is considered a critical food safety concern [15]. According to the literature, *Fusarium*, *Alternaria*, *Penicillium*, *Rhizopus*, and *Aspergillus* are the fungal genera that can grow in rice and produce secondary metabolites, such as mycotoxins, including OTA [16]. A plethora of studies reported the contamination of rice with OTAs in Portugal, Spain, Turkey, Egypt, Nigeria, Cote d’Ivoire, Morocco, Tunisia, Jordan, Chile, Vietnam, Japan, Korea, Italy, the United States, Iran, Bulgaria, the United Kingdom, and the Philippines [17,18].

In terms of rice consumption, the UAE and Lebanon reported 77 and 68.7 g per capita per day, respectively [19,20]. A precious study assessed the exposure to aflatoxin B1 from rice consumption in Lebanon [20]. To the best of our knowledge, no study has been conducted in Lebanon or the UAE to assess the safety of packed unprocessed rice in terms of OTA, or to determine the risk associated with its consumption. Therefore, the aim of our study is to measure the OTA in the packed unprocessed rice marketed in both countries, and to evaluate the exposure levels to this toxin from consuming this staple product.

## 2. Materials and Methods

### 2.1. Rice Sample Collection

The Lebanese and UAE markets were screened for different brands of rice, including white, parboiled, and brown. In Lebanon, 62 brands were identified, out of which 44 were procured twice in March and June 2021, while the remaining 17 brands were present in the market at either one of both collections, but not both. In the UAE, 89 brands were identified, out of which 38 were collected twice in March and June 2021, while the remaining 51 brands were found in the market at either of both collections. The total number of rice bags collected from Lebanon and the UAE were 105 and 127, respectively. The rice bags were stored at −18 °C until analysis was performed. To confirm whether or not a rice-packing facility had a food safety management system certification, the package was checked, and if no information was available in this regard, the facility was contacted by phone or email to inquire about the certification.

### 2.2. OTA Determination

The determination of OTA in rice was carried out using the enzyme-linked immunosorbent assay (ELISA) technique. This technique has been used to determine the OTA in cereals in a plethora of studies, for example, in Iran [21], the Czech Republic [22], Turkey [23,24], and Tunisia [25]. For this, the RIDASCREEN OTA 30/15R1312 test kit (R-biopharm, Darmstadt, Germany) was used. The test kit has 96 wells. As per [21,25], rice samples were spiked with OTA (5 and 10 ng/g), and each concentration was tested in 3 replicates. The average recoveries were 84% and 88% for the low and high concentrations, respectively, which were similar to those reported in the two aforementioned studies. The relative standard deviation (RSD) was calculated, and the values were lower than 10%, which are comparable to those reported by [26]. This proves the effectiveness of this kit in determining the OTA in rice.

Wells were added to the holder. Then, 50 μL of the standard or sample was pipetted into separate wells. Next, 50 μL of enzyme conjugate was added, and the plate was gently shaken to mix the reagents, and it was then incubated for 30 min at room temperature in a dark place. After that, liquid was poured out of the wells, and 250 μL of washing buffer was added to the wells. In addition, 100 μL of substrate/chromogen was added to each well. Then, the plate was manually shaken and incubated for 15 min at room temperature (20–25 °C) in the dark. Finally, each well was loaded with 100 μL of stop solution to stop any reaction. The absorbance was measured at 450 nm within 15 min of adding the stop solution by a plate spectrophotometer.

The estimation of the OTA concentration is based on plotting the standard curve, which is developed based on the absorbance values of the OTA standards (0, 0.03, 0.1, 0.3, 1, and 3 μg/L). The OTA concentration in μg/kg corresponding to the absorbance of each sample can be read from the calibration curve using the cubic spline function of the RIDA^®^SOFT Win (Art. No. Z9999) software. The limit of quantitation (LOQ) was 0.8 μg/kg, as reported by the kit manufacturer.

### 2.3. Moisture Determination

Rice was ground, and 2 g were weighed into the crucible. Crucibles were placed in an air oven at 130 ± 3 °C for 1 h. The crucibles were then taken out of the oven and weighed as soon as they reached room temperature. The weight of the crucible with the dried sample was obtained. The moisture was then calculated from the weight difference of the rice (AOAC, 1980, 22.013).

### 2.4. Exposure-to-OTA Assessment

The exposure level to OTA from rice consumption can be calculated by multiplying the average OTA concentration detected in the collected samples by the average rice consumption and then dividing it by the average body weight, as follows [27]:$$\mathrm{Exposure}\;\left(\mathrm{ng}/\mathrm{kg}\;\mathrm{body}\;\mathrm{weight}/\mathrm{day}\right)=\;\frac{\mathrm{Contamination}\;\mathrm{level}\;\left(\mathrm{ng}/g\right)\times \;\mathrm{amount}\;\mathrm{consumed}\;\left(g/\mathrm{day}\right)}{\mathrm{Body}\;\mathrm{weight}\;\left(\mathrm{kg}\right)}$$

The average daily consumption per capita of rice and average body weight in Lebanon are 68.7 g and 70 kg, respectively [20]. The average daily consumption per capita of rice and average body weight in the UAE are 77 g and 76 kg, respectively [19,28].

According to a recent EFSA report [29], the provisional tolerable weekly intake (PTWI) of 120 ng/kg bw/week previously set by EFSA is no longer valid for OTA. It was replaced by the margin-of-exposure (MOE) approach for the risk assessment of neoplastic effects and non-neoplastic effects. The MOE is the ratio between the benchmark dose level (BMDL10), which causes a 10% increase in cancer incidence in animals, and the daily dietary exposure. For non-neoplastic effects, a BMDL10 of 4.73 µg/kg bw/day was used, and an MOE below 200 indicates a health risk. For neoplastic effects, a BMDL10 of 14.50 µg/kg bw/day was used, and the MOE ratio should be above 10,000 [29].

### 2.5. Statistical Analysis

The OTA concentration was determined as the mean of 2 replicates. Data were entered into SPSS V27 for analysis. Testing for the normal distribution of the OTA concentration reported a strong positive skew in the data that was due to two large values that were deemed to be outliers and were deleted from analysis (the two values had an OTA concentration above 1). After removing the outliers, the OTA concentration was shown to have a normal distribution and, hence, was analyzed using parametric techniques. Means and standard deviations were used to evaluate the central tendency and measure of spread. The difference in the means between the groups was tested using the independent t-test for two groups, and the one-way ANOVA for more than two groups. When the ANOVA test showed statistical significance, a post hoc analysis was performed using the Bonferroni correction for pair-wise comparisons, which corrects the family-wise type I error. A significance level of 0.05 was used.

## 3. Results and Discussion

The overall average concentrations ± standard deviations of the OTA in the positive (above LOQ) rice samples were 1.29 ± 0.32 and 1.40 ± 0.42 μg/kg in Lebanon and the UAE, respectively. The contamination ranged between 0.8 (LOQ) and 4.98 in Lebanon, and between 0.8 (LOQ) and 9.58 μg/kg in the UAE (Table 1). Only one sample (1%) in Lebanon had an OTA level at the borderline of the European Union (EU) limit, and two samples (1.6%) in the UAE had an OTA level above the EU limit (5 μg/kg). All rice samples from Collections 1 and 2 had a moisture content <14% [30].

The OTA in brown rice was higher than in white and parboiled rice for both countries, yet the difference was not significant (*p* > 0.05). The packing season, packing country, and country of origin did not have any significant effect. The presence of a food safety management certification resulted in lower OTA in the rice, but the difference was significant (*p* = 0.03) in the UAE only, while a significant difference was found between both collections for the same brands (*p* < 0.05) in both countries. A longer time between packing and purchasing implied higher OTA levels, yet the effect was significant (*p* = 0.03) only in the UAE. Long grains had higher OTA than short grains, yet the difference was only significant in Lebanon (*p* = 0.001) (Table 2 and Table 3 for the effects of different independent variables on the OTA in rice in the UAE and Lebanon, respectively).

To the best of our knowledge, our study is the first to evaluate the safety of packed unprocessed rice marketed in Lebanon and the UAE in terms of the OTA, and to determine the exposure to this toxin from consuming this staple product.

The OTA levels in the rice samples in Lebanon (range: LOQ–4.98 μg/kg; mean: 1.29 ± 0.32 μg/kg) and in the UAE (range: LOQ–9.58 μg/kg; mean: 1.40 ± 0.42) were higher than in some studies but lower than in others. In Brazil, for example, in the period 2007–2009, the OTA concentrations in the rice ranged from 0.20 to 0.24 μg/kg [31]. OTA was not found in any of the rice samples collected in Ecuador [7]. In Portugal, the OTA contamination ranged from 0.09 to 3.52 μg/kg [32]. The OTA contamination in rice ranged from 0.08 to 47 μg/kg in Morocco [33], and from 0.65 to 11.54 μg/kg in Iran [34]. In Vietnam, Spain, and the United Kingdom, high amounts of OTA were reported, with levels ranging from 21.3 to 26.2 μg/kg, from 4.3 to 27.3 μg/kg, and from 1.0 to 19.0 μg/kg, respectively [35,36,37]. Nonetheless, in Lebanon, liquid chromatography was used to determine the dietary exposure to OTA from a total diet study in an adult urban population, and to estimate the mean concentration of the OTA in the rice and rice-based products [38]. The rice and rice-based items had an estimated mean OTA concentration of 0.68 μg/kg. Furthermore, when compared with [38], the mean concentration of the OTA in our sample was higher. One probable explanation is that the authors of [38] examined rice products rather than raw rice alone. Their study was carried out in 2014, when the rice brands available in the Lebanese market were different from the ones we collected. In fact, Lebanon is currently witnessing an unprecedented economic and political crisis, which has resulted in the emergence of new rice brands, imported and locally packed. As for the moisture levels, they were below the LIBNOR maximum of 14% of the weight, which can explain the relatively low levels of OTA in our samples compared with other studies.

Our findings revealed that brown rice had a higher level of OTA than white and parboiled rice, and this was not significant in both countries. This can be attributed to the presence of bran in the case of brown rice. In a Brazilian study, the average OTA level was highest in the rice bran compared with other parts [31].

Rice packed in spring/summer (from March to September) or fall/winter (from September to March) did not show any significant difference in terms of the OTA. This could be due to the fact that different packing countries are located in different geographical areas and, therefore, they do not have the same weather conditions, as OTA production is associated with the hot and humid season. The rice from developing countries of origin (India, Pakistan, Thailand, China) showed higher OTA levels than from developed ones (the United States, Italy), yet the difference was not significant. The mycotoxin issue is often more problematic in developing countries, and particularly in Asia, where climatic conditions and agricultural and storage techniques are favorable to fungus growth and toxin generation [39].

In the present study, the OTA in the rice collected from facilities having a food safety management system (FSMS), such as ISO22000, HACCP, and FSSC22000, was lower in both countries, yet the difference was only significant in the UAE (*p* = 0.03). For this, it is crucial to establish an integrated system based on the HACCP approach, from field to consumer, in order to control the OTA so it does not exceed the limits set by the legislation. Following good agricultural, storage, manufacturing, and distribution practices is important to decrease, as much as possible, the level of OTA before packing rice by reputable industries [40,41].

In our samples, higher concentrations of OTA were found in long-grain rice compared with short-grain rice, and the difference was significant (*p* < 0.001) in Lebanon only. This higher level of OTA in long-grain rice might be attributed to the higher surface area of this category of rice, which might result in more molds and, thus, OTA production [20].

When evaluating the effect of the time between the packing and purchasing of rice, the OTA levels were higher as this time increased; however, this was significant only in the UAE (*p* = 0.03). This could be due to the poor barrier properties in the packages of some rice brands, in addition to the poor storage of the rice in some retailers, which would increase the influence of the storage conditions on the quality of the rice.

Our results reported a significant difference (*p* = 0.001) between the brands of both collections and the levels of OTA in Lebanon and the UAE. This difference could be attributed to the inconsistency in the manufacturing and packaging practices, which increases the likelihood of OTA contamination in rice. Poor temperature and humidity control during storage, and not discarding rice grains with symptoms of fungal contamination, can lead to a higher contamination level [41].

For OTA, the provisional tolerable weekly intake (PTWI) is no longer considered a safety factor because of the uncertainty of the mode of action of OTA for kidney carcinogenicity, and the margin of exposure (MOE) is used instead [29]. Accordingly, the MOE, or the ratio between the benchmark dose level (BMDL10) and the daily dietary exposure, were calculated for both neoplastic and non-neoplastic effects. The average OTA levels in our study were 1.29 μg/kg in Lebanon and 1.40 μg/kg in the UAE. As previously mentioned, the average body weights were 76 kg in the UAE and 70 kg in Lebanon [20,28], while the average consumption of rice is 68.7 g/capita/day in Lebanon and 77 g/capita/day in the UAE [19,20]. The projected daily OTA exposure is therefore 1.27 ng/kg body weight/day in Lebanon, and 1.42 ng/kg body weight/day in the UAE. No health risk was observed for non-neoplastic effects (kidney lesions), as the MOEs in Lebanon and the UAE were both above 200 (3724 and 3330, respectively). For neoplastic effects (kidney tumors), the MOE values were 11,417 and 10,211 (above 10,000), indicating no health risk in Lebanon or the UAE, respectively. To the best of our knowledge, no study has evaluated the Lebanese population’s exposure to OTA from rice consumption. Raad et al. [38] calculated an average dietary exposure to OTA of 4.28 ng/kg body weight/day in an adult urban population in Lebanon, but their study was based on rice products, and not rice only. The dietary OTA exposure from the rice in Spain (0.17 ng/kg b.w./day) was lower than what our study reported in Lebanon (1.27 ng/kg b.w./day) and the UAE (1.42 ng/kg b.w./day) [36]. In contrast, the level of the OTA exposure assessed in our study was lower than that reported in Pakistan (4.2 ng/kg body weight/day), but higher than that in Iran (0.62 ng/kg body weight/day) [21,42]. Although the results of different studies are valuable references, the comparisons should be evaluated with caution because the studies may differ when it comes to the methodology and model used to determine the dietary exposure, types of rice and rice products, and the consumption patterns that might differ between countries and over time. As a result, even if rice is a significant source of OTA in one country, this may not be the case in other countries. Even though the determined average daily exposure to OTA in our study is not exceptionally high, it is critical to keep the levels as low as possible to avoid the harmful effects of OTA that are associated with increased rice consumption.

The strengths of our study include that it is the first in Lebanon and the UAE to evaluate the safety of the packaged unprocessed rice sold in the countries in terms of the OTA content, and to determine the OTA exposure levels from rice consumption. In addition, we analyzed packed rice, as, in general, the residents of Lebanon and the UAE purchase packed rice. With regard to the limitations, the results contained some outliers. This could be attributed to the chance of false-positive or -negative results due to the insufficient blockage of the surface of the microtiter plate immobilized with antigen, and future experiments should be performed in triplicates to solve this. Furthermore, many brands had no FSMS information. We were unable to contact these companies and obtain clear responses because they were unwilling to cooperate.

## 4. Conclusions

Rice is one of the world’s most common staple food commodities. Rice contamination with OTA is common all over the world. According to our study, the OTA contamination of the rice sold in Lebanon and the UAE is not currently a significant public health risk. Nevertheless, monitoring and surveillance strategies must be carried out regularly to ensure that rice is within the OTA-contamination limits. On the customer level, it is recommended that they purchase packaged rice brands that are food safety management system certified, and from recognized food stores. In this regard, because FSSC22000 involves unannounced audits, it is more reliable than ISO22000. Furthermore, rice must be appropriately stored in households away from sources of heat and humidity to reduce the risk of OTA contamination. Future studies should analyze the OTA in unpacked rice to obtain a more holistic view of the contamination levels. In addition, because rice is not consumed raw, additional studies must assess the effects of rice handling, preparation, and cooking that can affect the levels of contamination.

## Figures and Tables

**Table 1 ijerph-19-11074-t001:** Mean, range, and average OTA in rice collected from Lebanon and UAE.

	Lebanon	UAE
Maximum (µg/kg)	4.98	9.58
Average (µg/kg)	1.29	2.76

**Table 2 ijerph-19-11074-t002:** Effects of different parameters on OTA in rice collected in UAE.

Variable	Mean of Positive Samples (Above LOQ) (µg/kg)	SD	*p*-Value
Rice Type			
White/Parboiled	1.43	0.29	
Brown	1.28	0.27	0.326
Packing Season (UAE as country of packing)			
Fall/Winter	1.39	0.3	
Spring/Summer	1.38	0.3	0.422
Country of Packing			
UAE	1.27	0.27	
Other countries	1.25	0.25	0.952
Collection			
First	1.74	0.33	
Second	1.29	0.35	0.001
Country of Origin			
Developing (India, Pakistan, Thailand, China)	1.46	0.32	
Developed (the United States, Italy)	1.44	0.3	0.399
Food Safety Management System			
Presence	1.21	0.22	
Absence/Information Not Available	1.64	0.31	0.03
Grain Size			
Long	1.56	0.31	
Short/Medium	1.48	0.26	0.165
Time Between Packing and Purchasing			
From 1 to 9 weeks	1.22	0.2	
From 10 to 19 weeks	1.37	0.27	
From 20 to 29 weeks	1.46	0.3	
30 weeks and above	1.61	0.24	0.03

**Table 3 ijerph-19-11074-t003:** Effects of different parameters on OTA in rice collected in Lebanon.

Variable	Mean of Positive Samples (Above LOQ) (µg/kg)	SD	*p*-Value
Rice Type			
White/Parboiled	1.24	0.25	
Brown	1.32	0.27	0.246
Packing Season (Lebanon as country of packing)			
Fall/Winter	1.42	0.29	
Spring/Summer	1.35	0.3	0.282
Country of Packing			
Lebanon	1.44	0.29	
Other countries	1.4	0.32	0.769
Collection			
First	1.65	0.43	
Second	1.08	0.22	<0.001
Country of Origin			
Developing (India, Pakistan, Thailand, China)	1.38	0.29	
Developed (the United States, Italy)	1.23	0.32	
Not Available	1.36	0.28	0.221
Food Safety Management System			
Presence	1.41	0.31	
Absence/Information Not Available	1.34	0.34	0.51
Grain Size			
Long	1.48	0.32	
Short/Medium	1.18	0.21	<0.001
Time Between Packing and Purchasing			
From 1 to 9 weeks	1.16	0.24	
From 10 to 19 weeks	1.23	0.2	
From 20 to 29 weeks	1.48	0.39	
30 weeks and above	1.41	0.36	0.146

## Data Availability

The data presented in this study are available on request from the corresponding author.

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
