# Peer review of "Exposure to Ochratoxin A from Rice Consumption in Lebanon and United Arab Emirates: A Comparative Study"

_ijerph, 2022, doi:10.3390/ijerph191711074_

Round 1

Reviewer 1 Report

The study assesses OTA in rice from Lebanon and UAE, and calculates the exposure to this toxin from rice consumption in both countries. The study is highly needed as after looking at the literature, such work has not been done previously. The findings of this study can help both the policy makers and consumers to have a safer rice supply. Here are my comments about the work:

-          Please specify the classification of IARC for the OTA (L50-56)

-          I cannot see which studies used ELISA to determine OTA in cereals. Please list those studies in section 2.2

-          How did authors collect info on the presence of a food safety management system?

-          The authors reported measuring moisture content in section 2.3. Why did not they measure water activity instead?

-          I suggest that authors elaborate in the “conclusion” on what consumers need to do on household level to mitigate the OTA presence in their rice. For example, avoid unpacked rice? The authors stated “appropriately stored”, what is appropriate storage?

-          In section 4 “conclusion”, future studies must assess the effect of rice preparation and cooking on OTA contamination, especially that in Lebanon, rice cooking may differ from other countries

-          In the “conclusion”, consumers are recommended to buy brands coming from FSMS certified sources. Are all FSMS schemes equally reliable? ISO22000 vs. FSSC22000 for instance? Please elaborate on this.

Author Response

-          Please specify the classification of IARC for the OTA (L50-56)

Thank you for the comment. We added the following statement in Lines 54-56: "Presently, OTA is classified as a possible human carcinogen (group 2B) by the International Association for Research on Cancer (IARC) despite the fact that many researchers proposed to classify it as 2A instead (40)."

-          I cannot see which studies used ELISA to determine OTA in cereals. Please list those studies in section 2.2

Thank you for the comment. This comment overlaps with another comment by the other two reviewers. We added the following statement in Lines 95-96: "This technique was used to determine OTA in cereals in plethora of studies, for example in Iran (34), Czech Republic (41), Turkey (42, 43), and Tunisia (39)."

-          How did authors collect info on the presence of a food safety management system?

Thank you for the comment. We added the following statement in Lines 89-92: To confirm whether or not a rice packing facility has a food safety management system certification, the package was checked, and if no information was available in this regard, the facility was contacted by phone or by email to inquire about the certification.

-          The authors reported measuring moisture content in section 2.3. Why did not they measure water activity instead?

Thank you for the comment. Actually, the plan was to measure water activity among rice samples which show a moisture content above the limit (14%), but since all rice samples did not exceed the limit, water activity was not measured.

-          I suggest that authors elaborate in the “conclusion” on what consumers need to do on household level to mitigate the OTA presence in their rice. For example, avoid unpacked rice? The authors stated “appropriately stored”, what is appropriate storage?

Thank you for the comment. We added the following statement in Line 282: "away from sources of heat and humidity." The statement is now: "rice must be appropriately stored in households away from sources of heat and humidity to reduce the risk of OTA contamination."

-          In section 4 “conclusion”, future studies must assess the effect of rice preparation and cooking on OTA contamination, especially that in Lebanon, rice cooking may differ from other countries

Thank you for the comment. We added the following statement in Line 284-286: "In addition, since rice is not consumed raw, additional studies must assess the effect of rice handling, preparation and cooking that can affect the levels of contamination.."

-          In the “conclusion”, consumers are recommended to buy brands coming from FSMS certified sources. Are all FSMS schemes equally reliable? ISO22000 vs. FSSC22000 for instance? Please elaborate on this

Thank you for the comment. We added the following statement in Line 280-281: "Since FSSC22000 involves unannounced audits, it is more reliable than ISO22000."

Reviewer 2 Report

Manuscript ID: ijerph-1850260

Title: Exposure to Ochratoxin A from Rice Consumption in Lebanon and United Arab Emirates: A Comparative Study Special Issue: Ensure Healthy Lives and Promote Wellbeing for All at All Ages

Hussein F. Hassan et al.

The study presents result of the investigation of Ochratoxin A (OTA) contamination in packed rice, marketed in Lebanon and United Arab Emirates (UAE). The authors evaluated the study results in terms of the impact of selected variables on the level of OTA (brown or white rice, long or short grains, packaging season/ country, country of origin / marketing, presence of a food safety management certification, time between packing and purchasing). Moreover, based on estimated human exposure to OTA through to rice consumption, margin of exposure was used as a risk metric. Study results could be valuable for food scientist and producers, as well as for legal authorities.

Considering the following comments, recommendation is a major revision.

The incidence of OTA contamination is an obligatory element of the results in case of the studies dealing with food safety.

Lines 91-93: Apart from the LOQ and results of a recovery test, method validation as a minimum must include information related to method precision (at least calculate RSD from the repeated determinations performed within recovery assay (if understood correctly, it is stated that recovery was tested on 3 replicates (line 91), but this should be writen in a more clear way).

Lines 177-178: it is not meaningful to compare mean concentration related to sample collection with the legal limit – the limit is used to evaluate compliance of an individual sample.  

Tables 1 and 2: Result should be presented in more details, including range (min-max) of concentrations, at least. 

Author Response

Lines 91-93: Apart from the LOQ and results of a recovery test, method validation as a minimum must include information related to method precision (at least calculate RSD from the repeated determinations performed within recovery assay (if understood correctly, it is stated that recovery was tested on 3 replicates (line 91), but this should be writen in a more clear way)

Thank you for the comment. This comment overlap with another comment by another reviewer.
For the spiked samples with 5 and 10 ng/g of OTA, RSD of the recovery % were less than 10%, which was comparable to that of Maggira et al. (2022). The following statement was added to L101-102: "Relative Standard Deviation (RSD) was calculated and values were lower than 10%, which are comparable to those reported by (44)."

Also, we added “and each concentration was tested in 3 replicates” in L99. In addition, we added “for the low and high concentration” to make the statement more clear.

Lines 177-178: it is not meaningful to compare mean concentration related to sample collection with the legal limit – the limit is used to evaluate compliance of an individual sample. 

Thank you for the comment. The following statement was deleted: "Our investigation and that of (29) found that the mean concentration of OTA in rice is lower compared to EU limits (5 μg/kg)." (L186-187)

Tables 1 and 2: Result should be presented in more details, including range (min-max) of concentrations, at least.

Thank you for the comment. Although the range was mentioned in the text, we added a new Table: “Table 1. Range and average OTA in rice collected from Lebanon and UAE”

Reviewer 3 Report

This MS aims to determine the levels of OTA in packed rice in 2 countries (Lebanon and UAE). In addition, its aims to assess the exposure to OTA in both countries. While the MS is well written, it has several critical issues. 1) the rational to conduct study to is not clear? Why OTA and why in rice and why in both countries? Are there any connections or any preliminary data that support the objectives? The hypothesis is not clear to perform this study!!yet any scanning for mycotoxins is appreciated.

2) Method of analysis: based on the RIDASCREEN OTA website, the LOD is for rice is 0.8 mcg/kg which incomparable with the authors claim on the LOQ 0.015 mcg/kg. Does the author use any other instrument such as LC or LC/MS to compare? Is the method valid, if yes please include the reference, otherwise you have to go for whole method validation including sensitivity, specifity, accuracy, precision….etc. To our knowledge, it’s very challenging to achieve 0.015 mcg/kg by LC or LC/MS…how about ELISA!!!, how about the dilution factor and the reporting limit? I unable to see the full set of sample preparation and if the authors used any SPE to concentrate the sample. Sample size, this size of sample could be useful as input for full study, so am cautious about the sample size and the sampling over the different seasons. One of the bottleneck problems in analysis for mycotoxins is the sampling and the method of analysis? Have you run any QC samples along with your samples (preparation blank, calibration blank, LCS, check standards…..etc).

Assessment of exposure:

Not accurate references have been used as input for the average consumption of rice in Lebanon and UAE, I would recommend updating the references Ref 17) is a “commercial site”

Ref 20) is about the non-communicable disease (NCD), no single word about rice Ref 25) it’s for different country

Author Response

The rational to conduct study to is not clear? Why OTA and why in rice and why in both countries? Are there any connections or any preliminary data that support the objectives?

Thank you for the comment. We indicated in the manuscript that "no study was conducted in Lebanon or UAE to assess the safety of packed unprocessed rice in terms of OTA and to determine the risk associated with its consumption." (L75-77). The reason why rice was selected is the fact that it is a staple product in both countries (L73-75). On the other hand, the reason why Lebanon and UAE were selected, in addition to the absence of data on mycotoxins in rice in both countries, was the fact that the involved researchers in the study are part of a research team, which happens to be composed of members from Lebanon and UAE. We had the option of publishing the results of Lebanon and UAE in separate manuscripts, but we thought that comparing results from a developing country (Lebanon) to a developed country (UAE) would be a nice addition to the literature. We already assessed the AFB1 and published one manuscript on rice from Lebanon in the Journal of Food Protection (https://pubmed.ncbi.nlm.nih.gov/35146523/) while the AFB1 in rice from UAE is currently under review in an international journal.

Method of analysis: based on the RIDASCREEN OTA website, the LOD is for rice is 0.8 mcg/kg which incomparable with the authors claim on the LOQ 0.015 mcg/kg. Does the author use any other instrument such as LC or LC/MS to compare? Is the method valid, if yes please include the reference, otherwise you have to go for whole method validation including sensitivity, specifity, accuracy, precision….etc. To our knowledge, it’s very challenging to achieve 0.015 mcg/kg by LC or LC/MS…how about ELISA!!!, how about the dilution factor and the reporting limit? I unable to see the full set of sample preparation and if the authors used any SPE to concentrate the sample. Sample size, this size of sample could be useful as input for full study, so am cautious about the sample size and the sampling over the different seasons. One of the bottleneck problems in analysis for mycotoxins is the sampling and the method of analysis? Have you run any QC samples along with your samples (preparation blank, calibration blank, LCS, check standards…..etc).

Thank you for the comment. This comment overlaps with a comment by another reviewer.

We calculated LOQ ourselves by calibration curve method (LQ=10σ/s, where σ is the standard deviation of blank response and s the slope of calibration (L117-119). This method was already used in the literature, for example by Cantú-Cornelio et al. (2016)’s study published in Food Control.

ELISA is widely used as a screening method. The following statement was added in L95-96: “This technique was used to determine OTA in cereals in plethora of studies, for example in Iran (34), Czech Republic (41), Turkey (42, 43), and Tunisia (39).”

The recovery rates (above 80%) and associated relative standard deviation values (below 10%) was comparable to studies in the literature using ELISA and HPLC, such as Maggira et al. (2022). For this, the following statement was added: "Relative Standard Deviation (RSD) was calculated and values were lower than 10%, which are comparable to those reported by (44)." (L100-102)

Studies in the literature that used ELISA to screen did not do validation beyond calculating the recovery rate and RSD (RSD was calculated and added to the manuscript based on another reviewer’s comment). Our aim was to screen OTA in rice, and not to validate the kit, since the latter was already done by the kit manufacturer (r-biopharm, Germany). The kit manufacturer protocol was followed in our study point by point. If the reviewer wants us to run samples over HPLC as well, unfortunately, our laboratory is not equipped with the machine and thus, we cannot perform analysis but using ELISA.

Not accurate references have been used as input for the average consumption of rice in Lebanon and UAE, I would recommend updating the references Ref 17) is a “commercial site”

Thank you for the comment. For reference 17, " https://www.helgilibrary.com/indicators/rice-consumption-per-capita/" was replaced by the original source by FAO " https://www.fao.org/faostat/en/#search/rice"

Ref 20) is about the non-communicable disease (NCD), no single word about rice Ref 25) it’s for different country

Thank you for the comment. Reference 20 (UAE Ministry of Health and Prevention, accessed on 28 December 2021, https://cdn.who.int/media/docs/default-source/ncds/ncd-surveillance/data-reporting/united-arab-emirates/uae-national-health-survey-report-2017-2018.pdf) states the average body weight in UAE which we used to calculate the exposure to OTA from rice consumption in UAE (L132-134)

Reference 25 belongs to Iran. We used to compare our results to that of Iran (OTA contamination in rice ranged from 0.08 to 47 μg/kg in Morocco (24) and from 0.65 to 11.54 μg/kg in Iran (25)) - L178-179.

Round 2

Reviewer 2 Report

Manuscript ID: ijerph-1850260-R1

Title: Exposure to Ochratoxin A from Rice Consumption in Lebanon and United Arab Emirates: A Comparative Study

Hussein Hassan et al.

Non-resolved issues:

The incidence of OTA contamination is an obligatory element of the results in case of the studies dealing with food safety. Related to that: the LOQ of the analytical method presented in the manuscript is not supported by the ELISA-kit manufacturer and the study does not presents any quality assurance data. Minimum method performance verification included recovery test conducted on much higher concentration levels compared to the stated LOQ and level meaningful taking into account the mean OTA amounts in the analyzed samples (new Table 1, as well as Tables 2 and 3). The LOQ issue could be a probable reason for the very serious issue of the incidence of OTA contamination.

Tables 12 and 23: Result should be presented in more details, including range (min-max) of concentrations, at least (referring to tables denoted by table numbers). Without shortcuts as done by the introduction of new Table 1. And, yes, ranges presented in new Table 1 are already given in the text – at least from that reason it should be obvious that something more is needed.

Author Response

Thank you for the valuable comments. Kindly find below our response.

The incidence of OTA contamination is an obligatory element of the results in case of the studies dealing with food safety. Related to that: the LOQ of the analytical method presented in the manuscript is not supported by the ELISA-kit manufacturer and the study does not presents any quality assurance data. Minimum method performance verification included recovery test conducted on much higher concentration levels compared to the stated LOQ and level meaningful taking into account the mean OTA amounts in the analyzed samples (new Table 1, as well as Tables 2 and 3). The LOQ issue could be a probable reason for the very serious issue of the incidence of OTA contamination.

Thank you for the comment. Based on the comment of the reviewer, we considered only positive values in our study, i.e. those above the LOQ reported by R-biopharm, the kit manufacturer (0.8 microgram/kg), in calculating means and standard deviations, in addition to exposure levels. Also, we re-ran statistical analysis to assess the effect of different independent variables on positive samples only. We re-calculated the MOE for both neoplastic and non-neoplastic effects. Manuscript text and tables were updated accordingly.

Tables 12 and 23: Result should be presented in more details, including range (min-max) of concentrations, at least (referring to tables denoted by table numbers). Without shortcuts as done by the introduction of new Table 1. And, yes, ranges presented in new Table 1 are already given in the text – at least from that reason it should be obvious that something more is needed.

Thank you for the comment. In light of referring to the LOQ of the kit manufacturer, range is now LOQ to 4.98 microgram/kg, and LOQ to 9.58 microgram/kg in Lebanon and UAE, respectively. The manuscript text and Table 1 were updated accordingly. Tables 2 and 3 were better introduced in the "Results and Discussion" section. 

Reviewer 3 Report

Thank you for the reply. The authors unable to provide a reasonable response to conduct this study and method for analysis and a strong base to calculate the human exposure to OTA in rice in 2 countries. I looked also for other “reputable” company for ELISA Kit and also some SPE along with HPLC, but unfortunately, I was unable to find any with LOQ you came up with!! This could be "a screening study" after matching the LOQ with the manufacturer guidelines, which I believe is the optimum. More importantly, method of analysis and the LOQ, sample size and the references that have been used to measure the exposure need to be revised. For example, https://www.fao.org/faostat/en/#search/rice is more about crops and livestock products. Overall, method of analysis, sample size and the input data for measuring the exposure need to be reconsidered. Thanks!!

Author Response

Thank you for the valuable comments. Kindly find the following our response to them

1. Method of analysis

Thank you for the comment. Based on the comment of the reviewer, we considered only positive values in our study, i.e. those above the LOQ reported by R-biopharm, the kit manufacturer (0.8 microgram/kg), in calculating means and standard deviations, in addition to exposure levels. Also, we re-ran statistical analysis to assess the effect of different independent variables on positive samples only. We re-calculated the MOE for both neoplastic and non-neoplastic effects. Manuscript text and tables were updated accordingly.

2.  Input data for measuring the exposure need to be reconsidered

Thank you for the comment. For Lebanon, daily intake of rice per capita, and average body weight were obtained from Hassan et al. (2022), who conducted an FFQ based study to assess rice consumption patters in Lebanon.

https://meridian.allenpress.com/jfp/article-abstract/85/6/938/477734/Aflatoxin-B1-in-Rice-Effects-of-Storage-Duration

For UAE, daily intake of rice per capita was obtained using a USDA report in 2019, which estimated rice consumption in UAE per year to be 770,000 MT: 

Accordingly, daily consumption per capita of rice in Lebanon and UAE were updated in the manuscript, and exposure was re-calculated accordingly.

In addition, UAE population is estimated to be around 10M as per the World Bank: 

As for the average body weight, it is 76kg as reported by the Ministry of Health and Prevention: https://cdn.who.int/media/docs/default-source/ncds/ncd-surveillance/data-reporting/united-arab-emirates/uae-national-health-survey-report-2017-2018.pdf 

Manuscript and references were updated accordingly

3. Sample size

Thank you for the comment. As indicated in the manuscript, we screened the supermarkets in UAE and Lebanon, and all identified brands were collected twice. In other words, sample size could not be increased as we took whatever available in the markets of both countries.
